

# Ultraviolet disinfection impacts the microbial community composition and function of treated wastewater effluent and the receiving urban river

Imrose Kauser, Mark Ciesielski and Rachel S. Poretsky

Department of Biological Sciences, University of Illinois at Chicago, Chicago, IL, United States of America

## ABSTRACT

**Background**. In the United States, an estimated 14,748 wastewater treatment plants (WWTPs) provide wastewater collection, treatment, and disposal service to more than 230 million people. The quality of treated wastewater is often assessed by the presence or absence of fecal indicator bacteria. UV disinfection of wastewater is a common final treatment step used by many wastewater treatment plants in order to reduce fecal coliform bacteria and other pathogens; however, its potential impacts on the total effluent bacterial community are seemingly varied. This is especially important given that urban WWTPs typically return treated effluent to coastal and riverine environments and thus are a major source of microorganisms, genes, and chemical compounds to these systems. Following rainfall, stormflow conditions can result in substantial increases to effluent flow into combined systems.

**Methods**. Here, we conducted a lab-scale UV disinfection on WWTP effluent using UV dosage of 100 mJ/cm$^2$ and monitored the active microbiome in UV-treated effluent and untreated effluent over the course of 48 h post-exposure using 16S rRNA sequencing. In addition, we simulated stormflow conditions with effluent UV-treated and untreated effluent additions to river water and compared the microbial communities to those in baseflow river water. We also tracked the functional profiles of genes involved in tetracycline resistance (*tetW)* and nitrification (*amoA*) in these microcosms using RT-qPCR.

**Results**. We showed that while some organisms, such as members of the Bacteroidetes, are inhibited by UV disinfection and overall diversity of the microbial community decreases following treatment, many organisms not only survive, but remain active. These include common WWTP-derived organisms such as *Comamonadaceae* and *Pseudomonas*. When combined with river water to mimic stormflow conditions, these organisms can persist in the environment and potentially enhance microbial functions such as nitrification and antibiotic resistance.

Corresponding author
Rachel S. Poretsky, microbe@uic.edu

## INTRODUCTION

Wastewater treatment plants (WWTP) treat residential and industrial waste and return effluent to natural systems. In the United States, ∼20% of regulated effluent released from

WWTPs enter water bodies that can be classified as effluent dominated, i.e., where effluent discharge comprises the majority of the flow (*Brooks, Riley & Taylor, 2006*). Rivers that flow through cities are often used as receiving bodies for WWTP effluent, which typically introduces nutrients, compounds of emerging concern, and microorganisms to these systems (*Abraham, 2011*). Assessing the effects of effluent discharge on receiving waterways is of considerable environmental consequence, especially in areas under the influence of high population pressure and stress to the health of freshwater systems. In particular, WWTP effluent can potentially impact microbial community diversity, structure, and metabolic potential. The effects of effluent discharge on nutrient loading (*Waiser, Tumber & Holm, 2011*), chemical loading (*Garcia-Armisen et al., 2005*; *Ramond et al., 2009*; *Schlüter et al., 2007*), eutrophication (*Gücker, Brauns & Pusch, 2006*), and microbial communities (*Chu et al., 2018*; *Drury, Rosi-Marshall & Kelly, 2013*; *Goñi Urriza et al., 1999*; *Price et al., 2018*) have been investigated and show far-reaching impacts for the dissemination of compounds, genes, and organisms. For example, in a recent study of two WWTPs in Wisconsin, USA, we estimated that $\sim 30 \times 10^{12}$ bacterial cells per day are released from each plant's effluent into Lake Michigan, despite removal of most bacterial biomass (*Chu et al., 2018*; *Petrovich et al., 2018*). Futhermore, the impact of effluent on receiving water bodies can be greater after rain events that increase discharge from WWTPs that handle stormwater (*Chaudhary et al., 2018*; *Meziti et al., 2016*). Despite this, the primary method for assessing WWTP discharge water quality in the United States continues to rely on measuring fecal indicator bacteria (FIB) and largely ignores other microorganisms, genes, and many chemical contaminants (*United States Environmental Protection Agency, Office of Water, 2018*).

Each stage of wastewater treatment has the potential to alter the microbial community from the influent to the final effluent (*Petrovich et al., 2018*). The final treatment method used in the WWTP is one of the major influences on the microbial community composition and activity of effluent discharge. Secondary treatment, which removes at least 85% of biological oxygen demand and total suspended solids from the influent wastewater, is the minimum level that must be achieved for discharges from all municipal WWTPs under the Clean Water Act. Tertiary treatment and disinfection using chemical (commonly chlorine, chloramine, or ozone) or physical (e.g., ultraviolet light) processes is used by nearly every major municipal WWTP; however, according to the EPA Clean Watersheds Needs Survey (*United States Environmental Protection Agency, Office of Water, 2009*), approximately 50% of the US population is serviced by municipal WWTPs that do not provide more than secondary treatment and release effluent that has not been disinfected into the environment. The number of WWTPs that employ post-secondary treatment, including disinfection, is projected to increase by 2028. UV disinfection primarily works by damaging dsDNA and forming toxic photooxidation by-products that kill or damage microorganisms prior to effluent discharge (*Liang et al., 2012*). It is possible that this reduction in microbial load also reduces the input of specialized genes that are involved in biodegradation processes and/or enriches the community in UV-tolerant organisms, thus shifting the metabolic potential and microbial community diversity in the environment. Indeed, there is some evidence that UV treatment modifies the bacterial community in wastewater

(*Kulkarni et al., 2018*) and can enrich for some antibiotic resistant bacteria and genes in effluent, while removing others (*Di Cesare et al., 2016*; *Guo, Yuan & Yang, 2013b*; *Narciso-da Rocha et al., 2018*). These previous studies focused on the microbial community composition, which includes active as well as inactive organisms, or specific functions such as antibiotic resistance.

Here, we examined the potential effects of UV disinfection on the active microbial community in wastewater effluent as well as its impacts on the receiving riverine community by targeting the 16S rRNA and multiple functional genes in the community RNA fraction. Unlike previous studies on UV disinfection that assessed functional changes using microbial cultivation after UV exposure with a focus on pathogens (*Di Cesare et al., 2016*; *Guo, Yuan & Yang, 2013b*; *Kulkarni et al., 2018*; *Narciso-da Rocha et al., 2018*), we monitored the active microbial community with 16S rRNA to make predictions about potential ecosystem-level impacts of disinfection based on microcosm incubations. We focused on effluent from the Terrence J. O'Brien Water Reclamation Plant, Chicago, IL, (abbreviated O'Brien WWTP from here on), which discharges into the Chicago River Waterways. Effluent from the O'Brien WWTP has previously been shown to impact water quality (in terms of nitrogen and phosphorus) and microinvertebrate composition (*Polls et al., 1980*) as well as microbial community composition (*Chaudhary et al., 2018*) in this system. Until recently, the Chicago area remained the largest municipality in the US that did not disinfect WWTP effluent prior to release into the environment, providing a unique opportunity to assess potential impacts of disinfection; disinfection of O'Brien WWTP effluent using UV treatment began in 2016. We carried out a lab-scale UV disinfection experiment prior to the implementation of this post-secondary treatment in order to evaluate how the effluent bacterial community changes after UV disinfection. We also compared mock stormflow and baseflow conditions in microcosms with effluent and river water to make predictions about how UV disinfection might impact the river community under these conditions. Despite extensive work studying the effects of disinfection on microbial communities in effluent (*Di Cesare et al., 2016*; *Guo, Yuan & Yang, 2013b*; *Kulkarni et al., 2018*; *Narciso-da Rocha et al., 2018*), comparatively little is known about how this impacts microbial community composition and functional potential in receiving waters. We used a combination of phylogenetic and functional-gene-based molecular approaches to investigate the composition and diversity of the effluent, the functional ecology of the effluent-receiving river, and the fate and persistence of bacteria subjected to UV disinfection. Shifts in the diversity and composition of the effluent community over 48 h from UV exposure were observed. We used both inferred functions and quantitative PCR (qPCR) of specific functional genes associated with nitrification (*amoA*) and antibiotic resistance (*tetW*) in order to understand potential functional and ecosystem-level implications of UV disinfection. We demonstrate that different microorganisms respond differently to UV exposure and many bacteria survive and persist even after disinfection, including sewage specific *Arcobacter* as well as a variety of Beta- and Gammaproteobacteria. Our results can be used to predict the environmental implications of full-scale disinfection at the O'Brien WWTP as well as shed some light on the effects of this widely used disinfection process.

## MATERIALS & METHODS

### Site and sample description

The O'Brien WWTP on the North Shore Channel (NSC) of the Chicago River is one of the three largest WWTPs in the Chicago metropolitan area. The O'Brien WWTP has an average design flow of 333 million gallons per day (MGD) and a maximum of 450 MGD. It serves over 1.3 million people residing in $\sim$365 km$^2$, which includes the northern portion of Chicago and northern suburbs. It uses secondary treatment with waste-activated sludge processes and, at the time of this study, released an average of 0.787 million m$^3$ per day of treated but non-disinfected wastewater effluent into the NSC. The Chicago River system of channels and canals flows through a highly urbanized area with water inputs mainly from domestic pumpage and storm water runoff. According to US Environmental Protection Agency estimates, upwards of 70% of the Chicago River is comprised of wastewater and is often closer to 90% under stormflow conditions (*Illinois Department of Natural Resources, 2011*). O'Brien WWTP effluent and Chicago River samples (5–10 L) were collected in July 2014. Grab samples of the effluent from the WWTP discharge point and the river water 1 km downstream from the WWTP discharge point were collected using a horizontal sampler (Wildco, Yulee, FL, USA). All samples were stored on ice for transport back to the laboratory for subsequent experimental manipulations.

### Disinfection procedure and experimental manipulations

A bench-scale collimated beam apparatus design and dosage calculations were carried as described elsewhere (*Bolton & Linden, 2003*). The apparatus contained a monochromatic low-pressure (15 W) UV lamp housed in a dark enclosure. Effluent (1 L) was put under the collimated beam and gently stirred throughout the UV exposure time, which corresponded to a UV dosage of 100 mJ/cm$^2$. This fluence was chosen because it exceeds the municipality's standard requirements (*Metropolitan Water Reclamation District of Greater Chicago, 2011*) and is similar to the minimum recommended UV dose for the treatment of drinking water in the United States (*Linden et al., 2002*). Replicates of 100 mL microcosms with the UV-treated effluent or the untreated effluent were simultaneosuly incubated in the dark at room temperature (25 $\pm$ 2 °C) with gentle agitation (<200 rpm). Two microcosms were sacrificed for nucleic acid extractions at each timepoint: 2 h, 24 h, and 48 h. To further assess environmental implications, 50 mL of either UV-treated effluent or untreated effluent were mixed with 50 mL of river water and incubated as above. Unamended river samples reflect the river under baseflow conditions, where WWTP effluent contributes to $\sim$70% of the flow. The 50 mL amendments represent stormflow conditions of close to 90% effluent flow.

### Filtration and RNA extraction

At each timepoint, water/effluent samples were pre-filtered using 1.7 $\mu$m glass fiber filters (Whatman, Pittsburgh, PA, USA) and cells were collected on 0.2 $\mu$m polycarbonate filters (EMD Millipore, Billerica, MA, USA). Filters were stored in $-$80 °C until RNA extraction. An organic extraction method was performed as follows: 1.15 mg/ml lysozyme in lysis buffer buffer (50 mM Tris-HCl, 40 mM EDTA, and 0.73 M sucrose) was added to the filters and

incubated at 37 °C for 30 min on a rotator. The lysates were subsequently incubated with 1% SDS and 10 mg/ml proteinase K for 2 h at 55 °C while rotating. RNA was extracted from lysate with acid phenol and chloroform, and isolated via ethanol precipitation followed by suspension in TE buffer. DNase treatment was performed using the RTS DNase kit (MoBio Laboratories, Carlsbad, CA, USA) following the manufacturer's instructions. RNA (500 ng–1 μg) was transcribed into cDNA with High Capacity RNA-to-cDNA kit (Life Technologies, Carlsbad, CA, USA) according to manufacturer's instructions.

## 16S rRNA amplicon sequencing

For amplicon sequencing of the small subunit ribosomal RNA (SSU rRNA) of bacteria, primers 27F (*Frank et al., 2008*), and 534R (*Jumpstart Consortium Human Microbiome Project Data Generation Working Group, 2012*) were used to target and amplify the V1-3 hypervariable region. PCR reactions were prepared with 12.5 μl Accuprime Supermix II (Life Technologies, Carlsbad, CA, USA), 500 nM final concentration of each primer, 10-50 ng of cDNA, and water was added to a final 25 μl volume. Thermal conditions for PCR were as follows: 95 °C for 5 min, followed by 28 cycles of 95 °C for 30 s, 56 °C for 30 s and 68 °C for 5 s. A final, 7-minute elongation step was performed at 68 °C. PCR product size was confirmed with 1% agarose gel. Paired-end amplicon sequencing (2 × 300 bp) was done at the UIC DNA Services laboratory using the Illumina MiSeq platform, which yielded 26,537–48,074 reads per sample. All sequences have been deposited in the Sequence Read Archive under accession number SRP153092.

## Bacterial composition and function predictions

The quality of reads was assessed using FastQC (*Andrews, 2012*) and reads were trimmed for low-quality regions and primers using Trimmomatic (*Bolger, Lohse & Usadel, 2014*). Filtering, chimera checking, clustering, and taxonomy assignment were conducted using the Quantitative Insights Into Microbial Ecology (QIIME, v1.8.0) (*Caporaso et al., 2010*). Although paired-end reads were obtained, these did not pair well, likely due to length variability in the 27F-534R region that results in assembly of shorter fragments but not longer ones. Because of this, further analysis was only performed on the trimmed forward reads. Forward reads were quality trimmed and chimeric sequences were identified and removed with UCHIME using the *de novo* method (*Edgar et al., 2011*). Sequences were binned into Operational Taxonomic Units (OTUs) using usearch v. 7.0.109 (default settings) and the OTU table was filtered by removing OTUs with <0.005% of the total number of sequences and with no more than 15% of the samples being represented by singletons. Taxonomy was assigned following the closed reference OTU method where reads were clustered at 97% identity to a pre-existing Greengenes reference database (v13.8). Phylogenetic Investigation of Communities by Reconstruction of Unobserved States (PICRUST) v. 1.1.3 (*Langille et al., 2013*) was used to predict functions from the 16S rRNA datasets.

## Statistical analyses

Permutational multivariate analysis of variance (PERMANOVAs) were carried out in R (Adonis function, vegan package v. 2.4-4) using Bray–Curtis OTU-based distance

matrices to test the effect of the factors of time, UV disinfection, and stormflow vs. baseflow-like conditions. DESeq2 analysis (*Love, Huber & Anders, 2014*) was carried out using code from the Phyloseq (*McMurdie & Holmes, 2013*) tutorial "Using Negative Binomial in Microbiome Differential Abundance Testing," including the calculation of geometric means prior to DESeq2 testing to account for zero values. This method was used to identify differential abundances of taxa between treatments and is well-suited to experiments with low replication (*Love, Huber & Anders, 2014*). One-way Analysis of Variances (ANOVA) were run to test the effect of treatment on diversity. Additionally, we used linear discriminant analysis effect size (LEfSe) (*Segata & Huttenhower, 2011*) to compare the estimated phylotypes and identify the most differentially abundant taxa between different treatments with a moderately stringent effect size threshold of 2 (*Segata et al., 2011*). Taxonomic and functional profiles were compared using Statistical Analysis of Metagenomic Profiles (STAMP) (*Parks et al., 2014*). ANOVA and Tukey's 'Honest Significant Difference' tests were used to evaluate the qPCR-based gene expression between samples using the TukeyHSD() function in R. Random Forest models were used for supervised learning (*Knights, Costello & Knight, 2011*) using the supervised_learning.py script in QIIME with 1,000 trees and 10-fold cross validation. All statistical analyses were assessed for significance using an alpha level of 0.05.

## Quantification of gene expression

For detailed functional analyses, we focused on ammonia oxidation and tetracycline resistance. Real-time PCR analyses were performed according to MIQE guidelines. RT-qPCR of the bacterial ammonia monooxygenase (*amoA)* gene was conducted using primers AmoA-1F and AmoA-2R (*Rotthauwe, Witzel & Liesack, 1997*) on a Bio-Rad CFX96 instrument. Each reaction was performed in triplicate in a final volume of 20 µl containing 10 µl *Power* SYBR green PCR master mix (Life Technologies, Carlsbad, CA), 0.5 µM final concentration of each primer, 2 µl of 1:4 diluted cDNA template, and RNAse-free water. PCR amplification was initiated at 95 °C for 30 s followed by 40 cycles of denaturation at 95 °C for 15 s, primer annealing at 53 °C for 30 s, extension at 72 °C for 1 min, and plate read. The product specificity was confirmed by melting curve analysis (60–98 °C, 0.5 °C per read, 30 s hold). Expression of the tetracycline resistance gene *tetW* was quantified using primers from (*Aminov, Garrigues-Jeanjean & Mackie, 2001*; *Walsh et al., 2011*). Thermal cycling was as described above but with an annealing temperature of 64 °C. Transcript levels of all the genes were calculated by relative quantification using the $\Delta \Delta CT$ method (*Livak & Schmittgen, 2001*), with *rpoB* gene as the normalizing gene (*Dahllof, Baillie & Kjelleberg, 2000*). Cq values were converted to numerical values using the following formula: $\text{Log } 2^{-(\text{mean Cq}^{\text{rpoB}} - \text{mean Cq target gene})}$.

## RESULTS

### Effect of disinfection of effluent on bacterial diversity

We analyzed the 16S rRNA composition in UV-disinfected and control effluent microcosms over 48 h in order to evaluate shifts in the active microbial community in response to disinfection. We used this RNA-based approach to account for DNA that might be present

but no longer viable following UV exposure; it should therefore reflect the active microbial response to treatment (*De Vrieze et al., 2018*). Alpha diversity was assessed in the context of both evenness (Shannon Index) and richness (observed species) and compared across both treatment and time using ANOVA. Samples all had between 225–358 distinct OTUs. Overall, the changes in alpha diversity were generally small with alpha diversity (Shannon Index) between 3.0–5.0 for all five treatments. As expected, UV treatment resulted in a decrease in observed OTUs and reduced microbial diversity measured in terms of Shannon diversity, relative to the untreated effluent (Fig. 1). This was particularly evident after 48 h, when alpha diversity in the untreated effluent increased from 24 h prior but did not change in the UV treated effluent. In fact, despite a decrease in observed OTUs by an average of 73 OTUs between 24 and 48 h, neither diversity metric changed significantly over time in the UV-treated samples, but both increased between the beginning of the experiment and 48 h for the non-treated effluent samples (non-parametric $t$-test $p = 0.045$, observed species and $p = 0.032$, Shannon). Furthermore, the overall diversity was lower in the UV-treated samples relative to the control, although this was not deemed significant. Compositional change was assessed based on Bray–Curtis distance and showed that the microbial communities in both the untreated and UV treated effluent samples changed over time, but in different ways (Fig. 2A). Specifically, the Bray–Curtis distances between treated and UV-treated effluent samples were different when all timepoints, including time 0, were considered together (PERMANOVA $p = 0.025$). Further, the differences between community composition were significant over time for both treated and untreated effluent, as well as between treated and untreated effluent at 24 h and 48 h (PERMANOVA $p = 0.001$). Random Forest models used for supervised learning demonstrated that whether the sample was UV treated or not was more predictive of the community composition (Ratio of baseline error to observed error = 5.45) than was time.

## Effect of disinfection on effluent bacterial community composition

In all effluent samples, Bacteroidetes and Proteobacteria were the dominant phyla, with Bacteriodetes, primarily characterized by the families Cytophagaceae and Flavobacteriaceae, decreasing in relative abundance over time in the UV-treated effluent. In the untreated effluent, Alphaproteobacteria increased and Betaproteobacteria decreased in relative abundance over time (Fig. 3). The dominant Betaproteobacteria were either unclassified (∼16% of total OTUs) or members of the families *Comamonadaceae* (∼20%) and *Procabacteriaceae* (∼18%) (Fig. S1). Other abundant families were *Verrucomicrobiaceae* (∼5%), members of the Bacteroidetes *Flavobacteriaceae* (∼7%), ACK-M1 (∼7%), and *Cytophagaceae* (∼5%) (Fig. S1). *Pelagibacteraceae* were the most abundant alphaproteobacterial family (∼3%) (Fig. S1).

In order to determine which taxa were most characteristic of the differences between the untreated and UV-treated effluent (all timepoints combined), we used LDA Effect Size (LEfSe). Many OTUs decreased in relative abundance in the UV-treated effluent compared to the untreated effluent samples. These included an OTU most closely associated with the *Sediminibacterium* genus, relatives of which are common in freshwater and engineered systems such as activated sludge (*Ayarza, Figuerola & Erijman, 2014*),
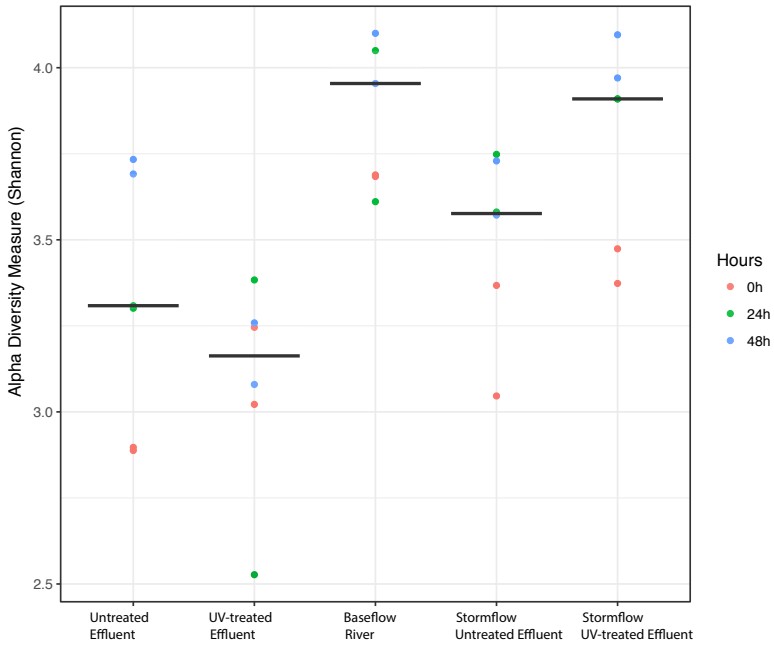

**Figure 1 Alpha diversity (Shannon diversity index) among the five experimental treatments.** Alpha diversity (Shannon diversity index) among the five experimental treatments. The diversity at 0 h (red), 24 h (green), and 48 h (blue) included for each condition with two replicates per time point. Stormflow samples indicate effluent additions to river water. The bold lines indicate median values for the six samples from each treatment.

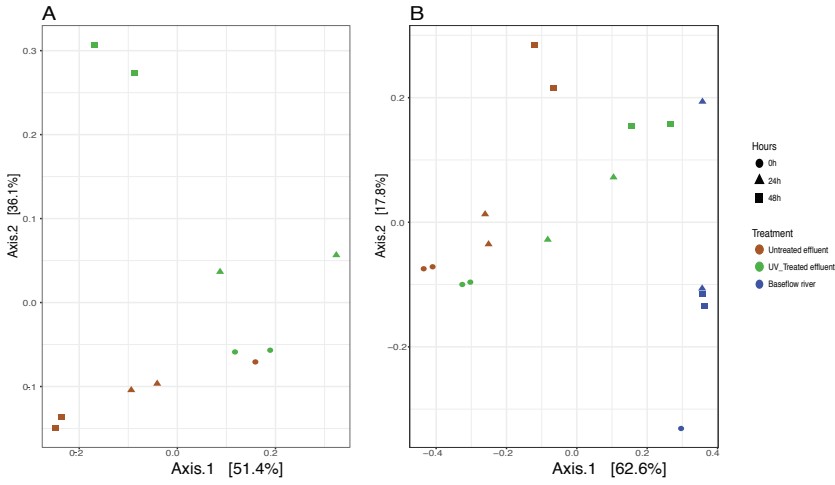

**Figure 2 Principle coordinates analysis (PCoA) ordination on Bray–Curtis distances of microbial communities.** (A) untreated (red) and UV-treated (green) effluent-only microcosms and (B) baseflow river water (blue), stormwater-like samples with untreated effluent (red), and stormwater-like samples with UV-treated effluent (green) at 0 h (circles), 24 h (triangles), and 48 h (squares).

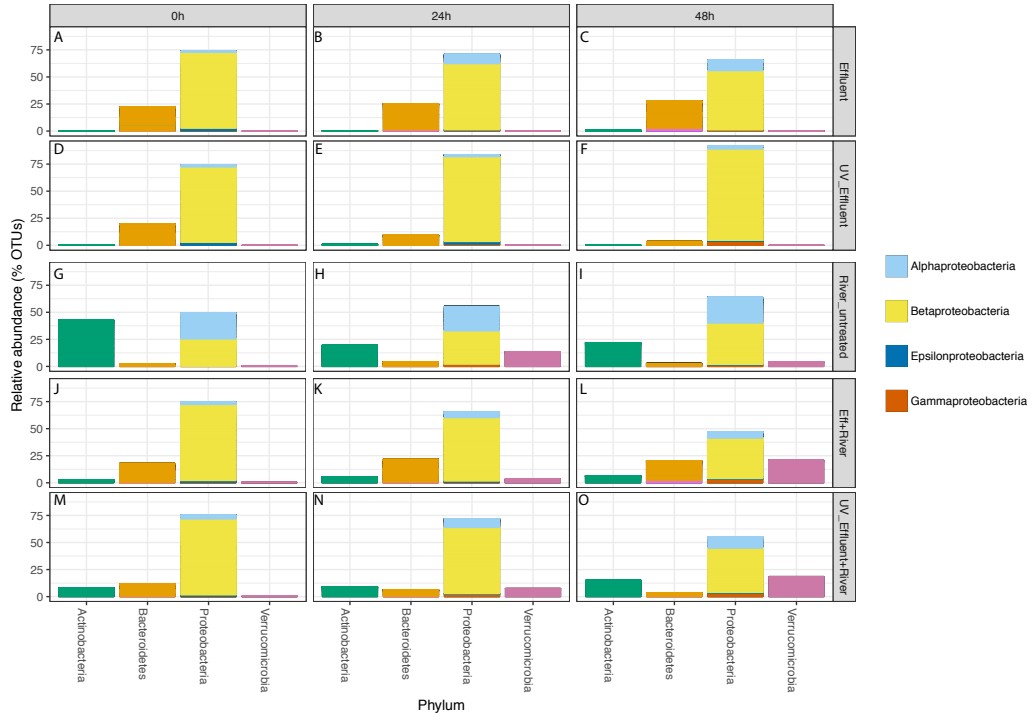

**Figure 3** **Taxonomic distribution of OTUs at the phylum level for the four phyla with a total of >1% of the OTUs in all samples.** Relative abundance refers to percentage of the OTUs attributed to each phylum with respect to all OTUs from each sample, including those that were unclassified. The Proteobacteria bars are subdivided into Alpha-, Beta-, Epsilon-, and Gammaproteobacteria. The five sample types are separated vertically by treatment (A–C) untreated effluent; (D–F) UV-treated effluent; (G–I) river water; (J–L) river with added untreated effluent; (M–O) river with added UV-treated effluent) and horizontally by time point (0 h, 24 h, 48 h).

as well as numerous OTUs affiliated with the *Rhodobacteraceae* and *Flavobacteriaceae* families. However, a number of organisms were significantly enriched following UV exposure. These included members of the Proteobacteria, families *Chromatiaceae* and *Moraxellaceae*, and genera most closely related to *Rheinheimera, Hydrogenophaga, Pseudomonas, Rhodoferax* (Fig. 4A). DeSeq2 analysis further identified OTUs belonging to the families *Comamonadaceae, Chromatiaceae, Pseudomonadaceae, Methylophilaceae, Rhodocyclaceae,* and *Procabacteriaceae* that were specifically enriched 48 h following UV exposure compared to the untreated effluent (Table S1). These same families significantly increased in abundance in the UV-exposed effluent over time (Table S1). By contrast, few OTUs changed in abundance over the course of the 48 h incubation in the untreated control effluent (Table S1).

In order to determine if the persistence of any organisms in the UV-treated effluent were fecal indicators, we examined the trends among organisms that are typically identified as coliforms and fecal enterococci, which include the genera *Enterobacter*, *Klebsiella*, *Citrobacter*, and *Escherichia* and other sewage indicator bacteria such as *Arcobacter* (*Fisher et al., 2014*), and compared their abundances to the untreated control effluent. Only 72

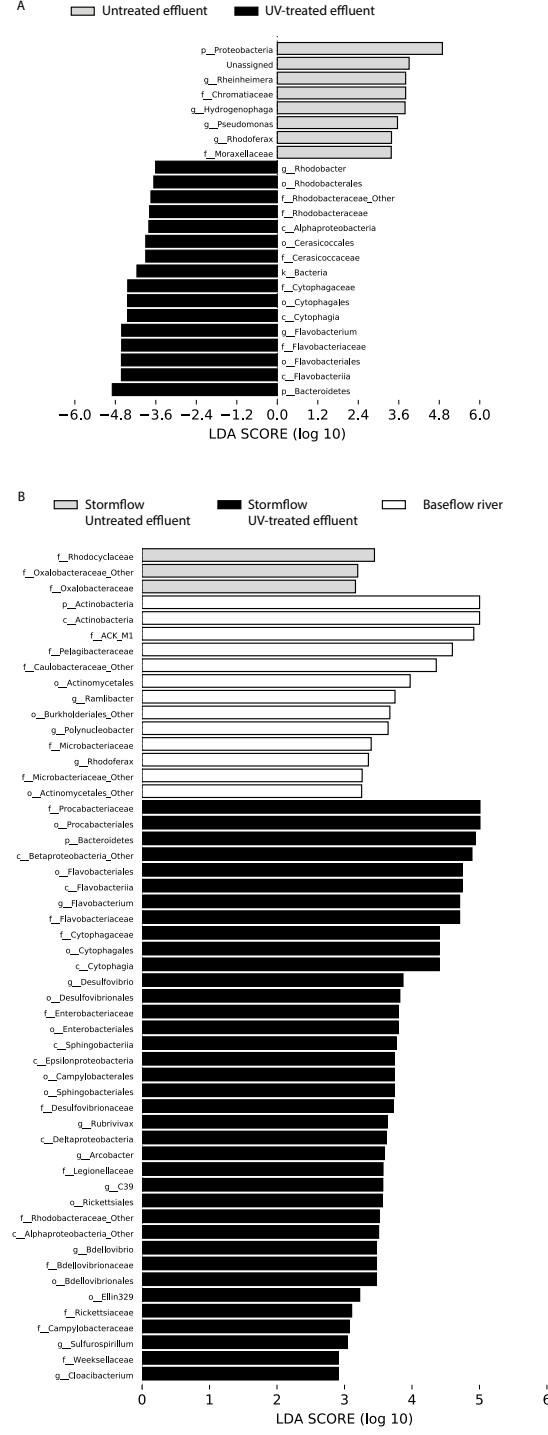

**Figure 4  LDA scores calculated by LEfSe of differentially abundant taxa.** (A) Untreated effluent (red) compared to UV-treated effluent samples (green) and (B) baseflow river (blue) compared to stormflow-like samples with untreated effluent (red) and stormflow-like samples with UV-treated effluent (green). All time points were combined for these analyses.

OTUs were assigned to taxa that matched these indicator bacteria: members of the orders *Sphingomonadales* (53) and *Enterobacteriales* (1), the genera *Dechloromonas* (1), *Arcobacter* (13), *Acinetobacter* (2), and *Legionella (2)*. Of these, only two *Sphingomonadales* that were between 5–15 times less abundant in the UV-treated than the untreated effluent were significantly different (all timepoints combined based on DeSeq analysis, $p = 0.000034$ and 0.011). Eleven OTUs affiliated with three *Arcobacter* OTUs and the two *Legionella* OTUs were actually more abundant in the UV-treated effluent samples, although these all generally decreased over time in the incubations in both conditions. This decrease, however, was not significant (Kruskall-Wallace test, $p = 0.84$ for *Legionella* OTU and 0.56 for *Arcobacter;* Table S2).

## Effect of UV disinfection on the river under stormflow conditions

Discharge of effluent from WWTPs is often a major source of stream-flow and chemical flux is many systems, but stormflow conditions can increase this WWTP-derived flow, thus impacting the microbial communities. In particular, WWTPs in the Chicago Area Waterways comprises more than 70% treated municipal wastewater effluent in baseflow conditions and up to 90% under stormflow conditions (USGA National Water Information System for North Shore Channel USGS 05536101 and *Illinois Department of Natural Resources, 2011*). Given the substantial influence of WWTP effluent in this system, we evaluated the impact of UV disinfection on the riverine microbial community into which it is discharged by combining either the UV-treated or untreated effluent with NSC river water at a ratio that mimics the ∼90% effluent stormflow. Although these microcosms do not necessarily reflect actual, system-wide effects, our observations allow us to make predictions about what might happen under stormflow conditions.

Despite the predominance of effluent in baseflow NSC river water, the river communities differed from the effluent communities in terms of both alpha diversity (Fig. 1) and composition (Table S1, Fig. 3), similar to what we observed previously (*Chaudhary et al., 2018*). The river samples had significantly higher alpha diversity (Shannon) than the effluent samples (non-parametric *t*-test $p = 0.04$). Proteobacteria and Bacteroidetes dominated both river and effluent samples, but river samples were also characterized by a high abundance of Actinobacteria (up to ∼13% of the river OTUs) and Verrucomicrobia (up to ∼10% of the river OTUs); both of these phyla contributed to <1% of the total effluent OTUs. Both phyla were primarily associated with the aquatic genera: *Prosthecobacter* and ACK-M1 (Figs. S1, S2).

The addition of effluent to river water, an approximation of stormflow conditions in the NSC, shifted the community compositions relative to the baseflow sample (river water only) immediately after effluent addition (Fig. 2B). The Bray–Curtis distances between baseflow and stormflow samples were significantly different when all timepoints were considered together (PERMANOVA $p = 0.003$), regardless of whether or not the effluent was UV-treated. In fact, there was no significant difference between the stormflow samples with UV-treated vs. untreated effluent addition (PERMANOVA $p = 0.102$). This similarity in overall community composition between the stormflow samples persisted over the course of the experiment with both stormflow treatments shifting in community composition

significantly over time (PERMANOVA $p = 0.001$) in the same way for both UV-treated effluent and untreated effluent stormflow samples (Fig. 2B). Only after 48 h did the community composition of two stormflow treatments begin to diverge from one another. The microbial community of the baseflow river samples did not change significantly over time (PERMANOVA $p = 0.067$).

LDA Effect Size (LEfSe) analysis identified several taxa that were most characteristic of the differences between the baseflow, untreated, and UV-treated effluent stormflow samples (all timepoints combined). Among the taxa that were more prevalent in the baseflow river water were members of the Actinobacteria as well as some common freshwater organisms including members of the families ACK-M1 and *Pelagibacteraceae* and the genus *Polynucleobacter* (Fig. 4B). Many taxa contributed significantly to differences in the stormflow samples with untreated effluent including fecal indicator members of the phylum Bacteroidetes, families *Enterobacteriaceae* and *Legionellaceae*, and genus *Arcobacter* (Fig. S2). The families *Rhodocyclaceae* and *Oxalobacteraceae* were the only groups driving differences in the UV-treated effluent stormflow water (Fig. S1).

At the end of the incubation experiment, DeSeq2 analysis showed similar taxa that were enriched in both stormflow treatments relative to the baseflow sample (Table S1). These included members of the families *Rhodocyclaceae, Cytophagaceae, Flavobacteriaceae, Verrucomicrobiaceae* and *Procabacteriaceae*. After 48 h, the UV-treated stormflow samples were also enriched in a *Campylobacteraceae* OTU whereas the untreated stormflow samples were enriched in a *Cryomorphaceae* OTU relative to baseflow. Interestingly, baseflow samples were enriched in an OTU attributed to *Pelagibacteraceae* relative to both stormflow samples. Only four OTUs were significantly different between the two stormflow treatments at 48 h; these included members of the families *Cryomorphaceae, Flavobacteriaceae,* and the order *Sphingobacteriales*, which were all more than twice as abundant in UV-treated compared to untreated effluent stormflow.

## Potential functional attributes

Based our previous observations of tetracycline resistance genes and ammonia oxidation genes in metagenomic datasets from both the O'Brien WWTP effluent and NSC river water (*Chaudhary et al., 2018*), we hypothesized that these functions could be affected by UV treatment. In addition, although the present 16S rRNA amplicon-based study focuses on microbial community composition rather than function, PICRUST analysis of the 16S rRNA datasets indicated possible differences in several functions, including antimicrobial resistance (more abundant in untreated effluent compared to UV-treated effluent, Welch's $t$-test $p = 0.045$, Fig. S3). We therefore used RT-qPCR to track the shifts in expression of a tetracycline resistance gene, *tetW*, and a bacterial ammonia oxidation gene, *amoA*, in order to evaluate if UV disinfection could change the expression levels of these genes and thus, whether there might be a potential for other functional shifts. *tetW* expression was significantly higher in the untreated effluent than in the UV-treated effluent (ANOVA $p = 0.0006$) (Fig. 5A). This same pattern was seen for bacterial *amoA* gene expression, although by 48 h *amoA* expression levels were no different between the effluents (Fig. 5A). Gene expression of both of these genes increased slightly over time in the effluents, although

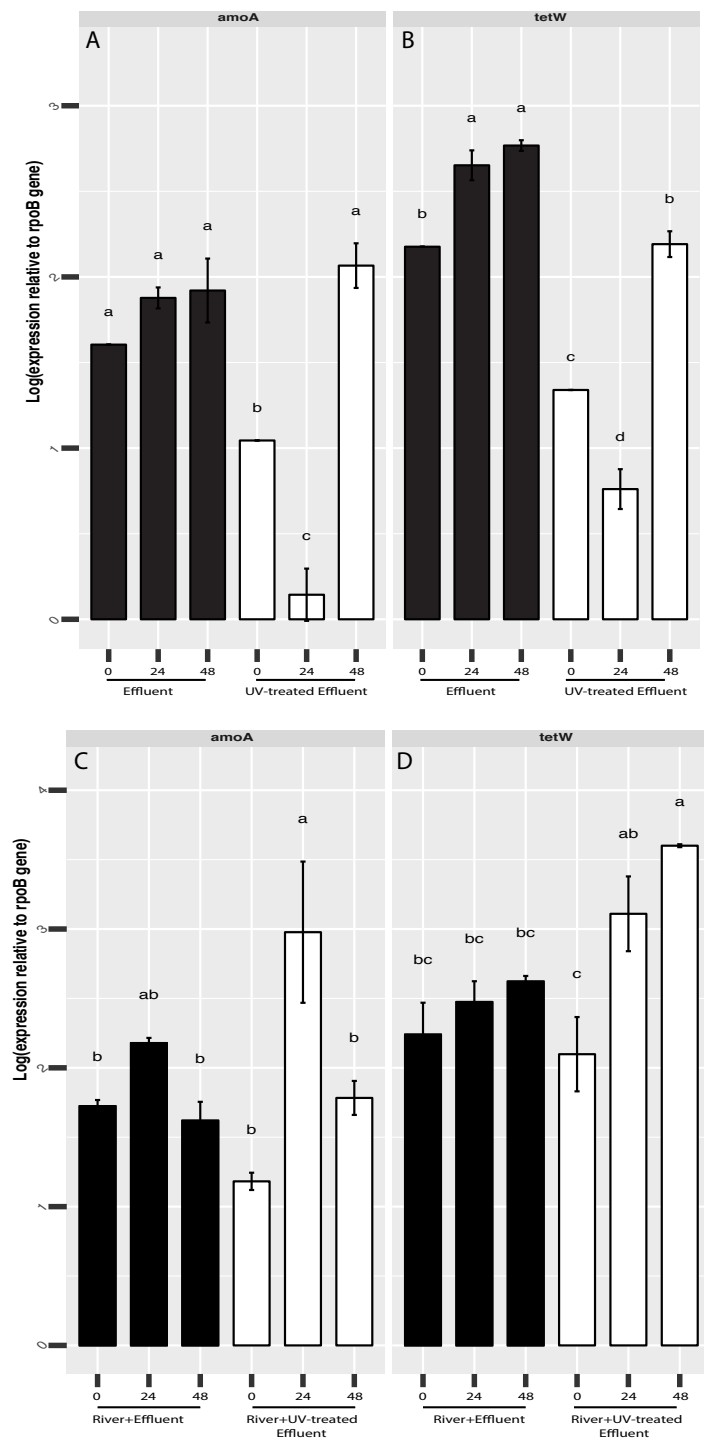

**Figure 5** **RT-qPCR-based quantification of *amoA* and *tetW* gene expression relative to *rpoB* gene expression derived from Cq values.** Expression of amoA (A) and tetW (B) in untreated (black) and UV-treated (white) effluent-only microcosms and amoA (C) and tetW (D) in stormwater-like samples with untreated effluent (black), and stormwater-like samples with UV-treated effluent (white) at 0 h, 24 h, and 48 h (two samples from each time point). Error bars indicate standard error from triplicate RT-qPCRs. Letters denote significantly different samples based on ANOVA and Tukey's 'Honest Significant Difference' tests.

this increase followed an initial decrease in the effluent samples exposed to UV. In contrast, *tetW* gene expression was higher in the river samples with UV-treated effluent (ANOVA $p = 0.016$) (Fig. 5B) and significantly increased in the river over time after the UV-treated effluent addition (Welch's *t*-test $p = 0.034$), but did not change over time in the river with untreated effluent (Fig. 5B). Bacterial *amoA* gene expression between river samples with both the untreated or UV-treated effluent was generally similar at all three timepoints.

## DISCUSSION

### A variety of bacteria survive and remain active in WWTP effluent following UV disinfection

UV treatment significantly altered the effluent bacterial community in our WWTP effluent samples. As a treatment designed to inactivate microorganisms (*Hijnen, Beerendonk & Medema, 2006*), UV disinfection indeed reduced the number of active OTUs and overall diversity (Shannon) in the effluent in our study. Although a recent report showed that UV treatment has little effect on microbial community composition in wastewater (*Narciso-da Rocha et al., 2018*), several others have shown reductions in both bacterial load (*Glady-Croue et al., 2018*), diversity (*Kulkarni et al., 2018*), and active/viable bacterial concentrations (*Hu et al., 2016*; *Sullivan et al., 2017*) following UV exposure of wastewater.

Organisms that have previously shown to be inactivated by UV treatment include *Aeromonas*, *Enterobacter,* and *Halomonas* (*Glady-Croue et al., 2018*; *Hu et al., 2016*; *Sullivan et al., 2017*), none of which we found to be major contributors to the effluent community here. Instead, we observed a substantial reduction in the relative abundance of Bacteroidetes OTUs, specifically *Cytophagaceae* and *Flavobacteriaceae*, following UV disinfection, which is notable as members of this is phylum dominates both sewage and, to an even greater extent, human fecal microbiomes (*Ahmed et al., 2017*; *Chu et al., 2018*; *McLellan et al., 2010*); however, we did not observe the typical sewage- and fecal-associated Bacteroidetes genus *Bacteroides* in our survey of the active community. In addition, we were unable to detect members of the *Lachnospiraceae* family, another sewage indicator group (*McLellan et al., 2013*), indicating that the WWTP used here was sufficient at either removing, inactivating these organisms, or decreasing their abundance substantially, even in the absence of disinfection. Therefore, the effects of UV treatment on effluent microbial communities are shaped by the initial community, which will vary between WWTPs based on treatment scheme and influent composition (*Shchegolkova et al., 2016*).

Some indicator bacteria (*Legionella* and *Arcobacter*) remained active following UV treatment and were more abundant in the disinfected effluent than the untreated effluent. The active fraction of the microbiome is therefore important in assessing effluent quality, as these are the organisms with the potential to persist in the environment following discharge. In addition to the two groups mentioned above, UV disinfection shifted the active community and increased the relative abundance of several organisms, mostly associated with Proteobacteria. Many of these, including *Comamonadaceae*, *Pseudomonas, Moraxellaceae,* and *Rhodocyclaceae* have previously been identified as among the most abundant taxa in sewage and freshwater (*Kulkarni et al., 2018*; *McLellan et al., 2010*;

*Narciso-da Rocha et al., 2018*; *Newton & McLellan, 2015*). *Rhodocyclaceae* in particular are common inhabitants of nutrient/substrate-rich environments such as wastewater and impacted urban streams (*Chaudhary et al., 2018*). *Comamonadaceae* are also abundant in freshwater environments (*Balmonte et al., 2016*; *Shaw et al., 2008*) and have previously been found to dominate in Lake Michigan (*Mueller-Spitz, Goetz & McLellan, 2009*), the freshwater source of the river we studied here. However, the OTUs affiliated with *Comamonadaceae* here were predominantly unclassified genera, rather than the common freshwater *Limnohabitans* (*Hahn et al., 2010*) and might instead be relative to WWTP-associated *Comamonadaceae* involved in denitrification that are common in activated sludge systems such as the WWTP from which we sampled (*Khan et al., 2002*).

Similar to what has been found in other wastewater surveys (*Ahmed et al., 2017*; *Chu et al., 2018*; *McLellan et al., 2010*), *Pseudomonas* was one of the common and dominant members here. This group is also known to tolerate and grow following UV treatment (*Glady-Croue et al., 2018*; *Hu et al., 2016*; *Sullivan et al., 2017*), which has been attributed to UV-inducible genes and UV-resistance plasmids that are often carried by members of this group (*Hu et al., 2016*; *Kokjohn & Miller, 1994*; *Zhao et al., 2018*). The other groups we saw active following UV treatment have not been implicated in UV tolerance in wastewater disinfection previously, but based on their abundances in the effluent studied here as well as in other WWTPs (*Shchegolkova et al., 2016*), their growth following UV treatment is notable. The *Moraxellaceae* family, in particular, includes the genus *Acinetobacter,* members of which can be either non-pathogenic or opportunistic pathogens (*Hare et al., 2012*) and are also among the predominant bacterial taxa in wastewater (*Ahmed et al., 2017*; *Chu et al., 2018*; *McLellan et al., 2010*). Some of the *Moraxellaceae* OTUs we saw increase in relative abundance following UV treatment were attributed to this genus, and previous work has demonstrated that several members of this group can survive UV exposure (*Hare et al., 2012*). In fact, we previously showed that *Moraxellaceae* were abundant in effluent from two different WWTPs, both of which employ disinfection (*Chu et al., 2018*). We therefore confirm the tolerance of several common wastewater microorganisms to UV disinfection at a standard UV dosage and reveal others whose activity post-UV exposure had not previously been documented.

## Stormflow derived from UV-treated effluent differs from that derived from untreated effluent

Despite the fact that WWTP effluent accounts for ∼70% of the river flow under base conditions in the system we studied, the river is still inhabited by many typical freshwater bacteria such as a variety of Actinobacteria including members of the ac1 clade of actinomycetes, freshwater *Pelagibacter*, and *Polynucleobacter* (*Hahn et al., 2011*; *Newton et al., 2011*; *Oh et al., 2011*). These organisms might originate from Lake Michigan, the freshwater source to the Chicago River. We previously observed an increase in the relative abundance of numerous bacteria under stormflow conditions in this system, which coincided with more than double the flow of non-disinfected effluent from the WWTP (*Chaudhary et al., 2018*). Freshwater bacteria made up a greater proportion of the baseflow river community and decreased significantly under actual stormflow conditions

(*Chaudhary et al., 2018*), which is what we observed here in the simulated stormflow and baseflow microcosms. Among the most significant changes in microbial community composition previously examined was an increase in *Legionella* in stormflow compared to baseflow river samples (*Chaudhary et al., 2018*). Since that study was conducted, the O'Brien WWTP has implemented a UV disinfection process prior to effluent discharge into the river. Here, we saw a notable increase in the Verrucomicrobia *Prosthecobacter* over time in both stormflow treatments compared to the baseflow, indicating that this riverine organism might thrive on nutrients added with WWTP effluent (*Hedlund, Gosink & Staley, 1997*). Although the two stormflow sample types did not differ much from each other initially, by 48 h the microbial community compositions diverged significantly. As with the *in situ* study (*Chaudhary et al., 2018*), we observed an increase in the relative abundance of *Legionella* in stormflow samples with untreated effluent in our microcosms. *Legionella* might become enriched during the WWTP chain (*Kulkarni et al., 2018*). Many other bacteria were also over-represented in the untreated effluent-derived stormflow samples compared to those that received UV-treated effluent. Several of these were the same organisms that survived and proliferated in the effluent only samples, such as members of the Flavobacteria, *Arcobacter*, Bacteroidetes, *Sphingobacteriales, Cryomorphaceae,* and *Cytophagales.* Similarly, *Rhodocyclaceae*, which was also found enriched in UV-treated effluent, was over-represented in the UV-treated effluent-derived stormflow samples. All of this indicates that the organisms that are released in WWTP effluent can proliferate in the receiving water body, including those that have survived UV treatment.

## Changes in the microbiome are reflected in expression of specific functional genes

Along with microorganisms, wastewater is a common source of antibiotics and antibiotic resistance genes to the environment, potentially creating an environmental hotspot and reservoir for antimicrobial resistance (*Barber et al., 2015*; *Chu et al., 2018*; *Mao et al., 2015*; *Rizzo et al., 2013*; *Tennstedt et al., 2003*; *Xu et al., 2015*). Although UV photolytic degradation of antibiotics can occur during disinfection and produce toxic photoproducts (*Dann & Hontela, 2011*; *Guo et al., 2013a*), bacteria susceptible to antibiotic photoproducts may obtain resistance by random mutations or acquire resistant via horizontal gene transfer, which could possibly be one of the reasons UV disinfection may shift the frequency of resistance genes in the effluent bacteria. In fact, our group has recently shown that several ARGs and ARBs persist through wastewater treatment with disinfection and these effluents are also enriched in mobile genetic elements (*Chu et al., 2018*; *Petrovich et al., 2018*).

The occurrence of ARB and ARGs in effluent presents a challenge to applying the UV disinfection process and conflicting results exist regarding its effectiveness at reducing ARB and ARG loads, which seems to vary with different antibiotics and treatment schemes. One study showed a reduction in ARBs following UV treatment (*Narciso-da Rocha et al., 2018*) and decrease in *mecA* and *vanA* ARGs after UV disinfection of wastewater was observed under laboratory conditions (*McKinney & Pruden, 2012*). In contrast, UV dose did not reduce the number of detectable *tet* gene types (tetracycline resistance) (*Auerbach, Seyfried & McMahon, 2007*) nor did UV disinfection contribute to significant reduction

of tetracycline- and sulfonamide-resistant bacteria concentrations in a full scale WWTP (*Munir, Wong & Xagoraraki, 2011*). More recently, several studies support these latter findings that UV disinfection does not reduce *tetW* genes and showed that it may actually increase the relative abundance of some ARGs and ARBs in effluent (*Glady-Croue et al., 2018*; *Guo, Yuan & Yang, 2013b*; *Hu et al., 2016*; *Sullivan et al., 2017*). Our results support these mixed findings and provide additional insight by evaluating gene expression for several days after UV treatment: expression of *tetW* decreased immediately following UV exposure compared to untreated effluent, but *tetW* expression increased in the river 48 h after the UV-treated effluent addition as compared with the addition of non-UV treated effluent. Concurrent with these results, the evidence of an increase in proteobacterial sequences, particularly *Pseudomonas*, may suggest that bacteria harboring antibiotic resistant genes following UV treatment also possess mobile genetic elements, which enable the proliferation of ARGs in the environment. Although we did not explore mobile elements here, previous studies indicate that mobile elements can be enriched during treatment and correlate with ARGs (*Chu et al., 2018*; *Hu et al., 2016*; *Petrovich et al., 2018*; *Wang et al., 2013*).

WWTP effluents are also a source of high levels of organic matter and nutrients, including ammonia (*Brion & Billen, 2000*; *Servais et al., 1999*) and are known to impact ammonia oxidizing microorganisms in receiving waters (*Carey & Migliaccio, 2009*; *Merbt et al., 2015*). Although UV treatment initially reduced the expression of *amoA* in effluent, expression levels were the similar at the end of the incubation period. Furthermore, *amoA* gene expression was similar in the stormflow samples with treated and untreated effluent. Taken together, our results suggest that like *tetW* gene expression, the bacteria carrying out ammonia oxidation are resilient to UV treatment 48 h after exposure. Photoinhibition (non-UV) of *amoA* has been documented previously (*Merbt et al., 2017*), but this is the first evaluation, to our knowledge, of nitrification activity in effluent following UV exposure. Given that both *amoA* and *tetW* gene expression recover to levels similar to those in untreated effluent within 48 h of UV treatment, it is likely that a wide variety of functions are resilient to UV treatment and can persist when introduced into the surrounding environment.

## CONCLUSIONS

UV exposure decreased the number of OTUs and the microbial diversity of effluent discharged from a WWTP that did not employ a disinfection step before discharge into an urban river. Several organisms remained active following UV exposure and were enriched relative to untreated effluent, including *Moraxellaceae, Pseudomonas, Comamonadaceae,* and *Rhodocyclaceae.* When potential ecosystem-level effects were considered, stormflow-like river samples with UV-treated effluent had fewer organisms like *Enterobacteriaceae, Legionellaceae, Arcobacter* compared to stormflow with untreated effluent. At a functional level, UV treatment initially decreased gene expression of both *tetW* and *amoA*, but these funtions recovered over time. Our study was based on a single sampling event at a single WWTP, so repetition would be helpful for determining if our findings are representative

of the plant over time or even of other WWTPs. Additional functional analysis using metagenomics or metaproteomics would also add a deeper understanding of UV effects on the microbial community. Despite these limitations, our comparison of UV-treated and non-UV treated effluent using lab-scale disinfection experiments provided insights into the effects of disinfection on the effluent total bacterial community and its implication on the environment.

## ACKNOWLEDGEMENTS

We thank the staff of the Metropolitan Water Reclamation District for facilitating sample collection. Juana Villagomez provided technical assistance as a Bridges 2 Baccalaureate summer student.

### Funding

Funding was provided by the University of Illinois Chicago start-up funds and a grant from the UIC Honors College. The funders had no role in study design, data collection and analysis, decision to publish, or preparation of the manuscript.

### Grant Disclosures

The following grant information was disclosed by the authors:
University of Illinois Chicago start-up funds and a grant from the UIC Honors College.

### Competing Interests

The authors declare there are no competing interests.

### Author Contributions

- Imrose Kauser conceived and designed the experiments, performed the experiments, analyzed the data.
- Mark Ciesielski performed the experiments.
- Rachel S. Poretsky conceived and designed the experiments, analyzed the data, contributed reagents/materials/analysis tools, prepared figures and/or tables, authored or reviewed drafts of the paper, approved the final draft.

### DNA Deposition

The following information was supplied regarding the deposition of DNA sequences:
All sequences are available in the NCBI Sequence Read Archive: SRP153092.

### Data Availability

The raw qPCR data Cq data for *rpoB*, *amoA* and *tetW* are available in Supplemental Files.

## Supplemental Information

Supplemental information for this article can be found online at http://dx.doi.org/10.7717/peerj.7455#supplemental-information.

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
