# Peer review of "Ultraviolet disinfection impacts the microbial community composition and function of treated wastewater effluent and the receiving urban river"

_PeerJ, doi:10.7717/peerj.7455_

## Round 0.1 · original submission · Minor Revisions

Thank you for your submission. I am pleased to indicate that both reviewers were senior scientists with expertise in wastewater community ecology and were complimentary of the rigour and clarity of your manuscript. Both expressed dismay that the study was so limited in scope, but conclusions were not overextended and this is appropriate subject matter and scope for PeerJ. I encourage the authors to do a careful revision of the manuscript attending to all of the recommendations of the reviewers, and we look forward to your revised draft.

Reviewer 1 ·

Basic reporting

The manuscript “ Ultraviolet disinfection impacts…” by Kauser et al. is well written. Intro and background are informative and rationale clearly stated. Literature reference appear to be valid and relevant throughout the manuscript. Reference section appears to be flawless which is rare these days. Structure of the manuscript confirms to PeerJ standards. No issues with fig 2—5, but it is hard to understand fig 1 without the details in the method or caption: two replicates are plotted for each time point?

Experimental design

Research is original and within Aims and Scope of PeerJ, but more detail is needed.
Lane 144-150. It was just one experiment run simultaneously? How many replicates where sacrificed at each time point? Two? (30 samples total?)
Lane 184. Only the forward reads were used?
Lane 197 DeSeq2…I’m not familiar with this program, what was it used for?
Lane 218. As SybrGreen based assay was used, melting curve analyses was not conducted?
I’m not sure if base and stormflow lab-setup (just based on the %) and conclusions will hold and reflect bacterial community dynamics in real-world situations.

Validity of the findings

Data and replication is limited but appear statistically sound
Conclusions are limited but valid based on the results

Additional comments

Authors have put quite a bit effort into writing this manuscript and clearly have a good knowledge of background research/literature It is just unfortunate that the paper appears to be based on the analyses of one wastewater effluent sample part of which was UV treated, both of which where then diluted or undiluted in river water and analyzed at 0, 24 and 48h (30 samples?).
Somewhat ambitious goal to assess potential ecosystem-level impact (lane 92-93), not sure if this can be deduced from a small lab experiment which involved mixing river water with sewage.
Concern is that only one experiment with limited data points (three), but authors do acknowledge weaknesses in the conclusions. Conclusions are limited but valid.

·

Basic reporting

The manuscript is well written, in an unambiguous, professional English.
The literature references is well documented with many recent publications. However, the introduction could better describe the state of the art in that particular field, and emphasize what has been done and what is missing. That would help the reader to understand what gap this paper is filling.
Some figures could be improved (see general comment).

Experimental design

Research questions are well defined, relevant and meaningful. As stated above, the introduction could be re-structured to be able to identify right away how this research fills an identified knowledge gap.
The experimental design seems appropriate to answer the question. However, details about the experimental design are missing (see general comments).

Validity of the findings

Duplicate of experiments was done. RT-PCR where run in triplicates.
Data seems robust. A minority of statistical tests are irrelevant (comparison of 2 vs 2 values or 3 vs 3 values), but that issue can be easily addressed (see the general comments for more details).
Conclusions are well stated, linked to the original research question and limited to supporting results.

Additional comments

General comment

In the present study, the authors performed a lab experiment to evaluate the impact of UV disinfection on the bacterial community composition and activity in the wastewater effluent and receiving waters. An RNA-based approach was used to avoid the amplification of free or damaged DNA, which is particularly relevant after wastewater treatment processes. Active bacterial community structure was assessed targeting the V1-V3 region of the 16S rRNA gene. Bacterial activity was monitored using quantitative RT-PCR of the bacterial ammonia monooxygenase (amoA), while resistance gene was assessed by targeting the tetracycline resistance gene (tetW).
The authors collected effluents from a wastewater treatment plant (WWTP) in Chicago (that did not undergo a tertiary treatment) and river water samples located 1 km downstream from the WWTP discharge point. Three conditions were tested: (1) to evaluate the influence of UV-disinfection on wastewater effluent, effluent samples were UV-treated or not. (2) To assess environmental implications, treated or untreated effluent samples were added to river samples to mimic stormwater conditions. (3) River sample no spiked with effluent was also monitored to mimic baseflow conditions.
The authors observed that while some organisms are inhibited by UV disinfection (alpha diversity decreases following treatment), many organisms not only survive, but remain active. When effluent were combined with river water, some organisms could persist in the environment and likely enhance microbial functions such as nitrification and antibiotic resistance.

Overall, the paper is well written and easy to read. Figures are appropriate and supplementary data provides useful information and data. I only have three concerns. First, in the Introduction, the authors should better describe the literature and the gaps that remain in the field (more RNA-based approach, etc.). Second, the Method section could be more precise. For example, details about the microcosm incubation conditions are missing (e.g., agitation speed, light, etc.). Moreover, the authors have to describe the statistical procedure they followed to perform the random forest regression analysis (package, number of trees, etc.). The version of usearch should be provided. Does the default LDA threshold of 2 is relevant in the context, a threshold set at 3 would be more stringent. Finally, although the results are congruent with in situ observations (Chaudhary et al. 2018), I would highlight the fact that results are originated from a laboratory experiment that might not entirely reflect what can be observed in situ.


Specific comments:

Abstract
Line 22: the authors already introduced “WWTP” in the text, so “urban wastewater treatment plants” could be deleted.
Lines 24-25 (and lines 61-62): that is true for combined systems.
Line 28: since a lot of studies used “16S rRNA” to describe DNA amplicons, I would write in the abstract the word “RNA” alone (somewhere) to make it clear that the authors targeted only RNA and not DNA (to highlight that specificity).

Introduction
Lines 48-50: redundant with the 2nd sentence.
Line 63: it is true for the US, but the monitoring of FIB to evaluate the effluent quality is not the case in all countries of the world.
Paragraph 66-85: I would reorganize this paragraph. The first sentence is not completely true: in the presence of a secondary and tertiary treatment, both steps must drastically shape the bacterial community composition (BCC) and activity (BCA) (and not only the final treatment). With all the studies made on this particular topic (see for example the next part of the paragraph or the beginning of the discussion), the authors can’t state that “the effect of disinfection on microbial community composition and functional potential in receiving waters is unknown”. If it is actually true, they have to better explain what have been done in the field, and what is missing (what they partially did in the next part of the paragraph).

Methods
The versions of the R packages or commands that the authors used should be mentioned. Some citations are missing in the “statistical analysis” section.
I understand that the authors don’t want to describe in details the Picrust analysis (for only three lines in the Results section), but this analysis should be introduced in the Method section.
Line 143: Does “Chicago 2011” refers to “Illinois Department of Resources 2011” or “Metropolitan Water Reclamation District of Greater Chicago 2011”?

Results
Please consider to not cite references in the Results sections when it is redundant with the Methods section.
Lines 237-243: Did the authors statistically tested two points against two points? I would just mention “the alpha diversity is lower or higher”, not much.
Moreover, I would specify that despite the difference visible in Figure 1, the alpha diversity is about the same between the five treatments (around 3.5).
Line 246: Does “all timepoints” include T0?
Lines 284-285: Something is wrong with the sentence, “identified that could be” should be deleted?
Line 286: How Arcobacter can be both an indicator bacteria and a genus identified to be associated with the indicator bacteria?
Line 289: Something is missing after “with”?
Line 290: Consider “OTUs” instead of “OTUS”
Lines 300-302: The parenthesis can be deleted without loose information since it has been clearly described previously.
Lines 315-317: Overall in that part, the authors keep explaining the treatment 2 (effluent + river samples = stormwater conditions) and treatment 3 (river sample = baseflow conditions). That makes the text hard to follow. By removing these repeats, the text would be more clear and concise.

Discussion
Lines 404-415: The authors highlighted the fact that some taxa predominant in effluent are also associated in the literature with natural habitats. If publicly databases are available (V1-V3), the authors could verify if the taxa they recovered are mainly associated with human fecal sources, sewage infrastructures or environmental sources.
Line 412: The authors mention that some OTUs couldn’t be taxonomically assigned to the genus. In that specific example, the use of the out-of-date Greengenes database can be challenged. The use of more recent databases could solve the issue.
Lines 424-433: In addition to Arcobacter spp., Acinetobacter spp., are also among the predominant bacterial taxa in sewage influent (see references already cited by the authors in the paper).
Line 438: Did the authors observed OTUs assigned to the acI lineage? I’m surprised to observe them in rivers since acI are mainly inhabitant of lacustrine environments (I can be wrong – but Chicago river is fueled by Michigan Lake?).
Line 448: Please consider “published” instead of “done”.
Line 449-450: I wouldn’t mention that detail in the manuscript.
Line 454: Please consider “significantly” instead of “signficantly”
Line 474: I would delete “HGT” since the authors never use the abbreviation after that.
Line 493: Does the word “significantly” refers to a statistical test comparing 3 points (24h) against 3 points (48h)? I would just say that the expression drastically dropped after 24h but reach the “control” level expression after 48h, as the authors did in the Results section.


Figures
Figure 1:
I would delete the sentence “Effluent samples are effluent only”.
I am not sure that boxplots are the most appropriate way to plot the data considering that there are only 6 points per box. Maybe a jitter plot or a pirate plot would be more appropriate.
I can’t see 6 points for each boxplot. Are they overlapping with other points or did the authors exclude some samples from the analysis/sequencing?
Figure 5:
The number of replicates should be mentioned in the legend. Moreover, I'm still not sure if a histogram is the most appropriate way to plot these data (see Figure 1).

Supplementary data
Standard curves are missing to fully interpret the RT-qPCR results.

---

## Round 0.2 · Minor Revisions

Please review and address the remaining minor suggestions made by Reviewer 2 and we think this will be accepted.

Reviewer 1 ·

Basic reporting

The manuscript is well written. Introduction and background are sufficient and informative. The rationale is clearly stated. Literature reference appear to be valid and relevant throughout the manuscript. Structure of the manuscript confirms to PeerJ standards. The concerns have been addressed.

Experimental design

Research is original and within Aims and Scope of PeerJ. Sufficient information has been provided in the methods section. The concerns have been addressed

Validity of the findings

Data and replications is limited but appears statistically sound. Data has been provided. The concerns have been addressed.

Additional comments

The concerns have been addressed.

·

Basic reporting

no comment

Experimental design

no comment

Validity of the findings

no comment

Additional comments

I really appreciated the effort the authors made to respond to our remarks and questions.
The authors took into account the suggestions I made. If not, they justified their position (I was not expecting such vivid argumentation, especially regarding the greengene database).

I have some minor comments that should not need to be reviewed before publishing the manuscript. I made a comment regarding the line 28 in the abstract. (1) Because of this comment, the sentence is off now. Sorry, the first version was better. (2) Add "RT-" before qPCR on line 32 (it was likely what I wanted to say the first time). (3) One space is missing before the parenthesis line 172. (4) There is a display problem line 262.

---

## Round 0.3 · accepted · Accept

Dear Dr. Poretsky,

Thank you for your submission. I have reviewed your revised manuscript and am happy to accept it for publication in PeerJ.